# The Effects of Family Financial Stress and Primary Caregivers’ Levels of Acculturation on Children’s Emotional and Behavioral Problems among Humanitarian Refugees in Australia

**DOI:** 10.3390/ijerph17082716

**Published:** 2020-04-15

**Authors:** Linlin Yu, Andre M.N. Renzaho, Lishuo Shi, Li Ling, Wen Chen

**Affiliations:** 1Department of Medical Statistics, School of Public Health, Sun Yat-sen University, Zhongshan Road 2, Guangzhou 510080, China; yullin@mail2.sysu.edu.cn (L.Y.); shilsh5@mail2.sysu.edu.cn (L.S.); lingli@mail.sysu.edu.cn (L.L.); 2Sun Yat-sen Centre for Migrant Health Policy, Sun Yat-sen University, Zhongshan Road 2, Guangzhou 510080, China; 3School of Social Science, Western Sydney University, Locked Bag 1797, Penrith, NSW 2751, Australia; Andre.Renzaho@westernsydney.edu.au; 4Translational Health Research Institute, Penrith, NSW 2751, Australia

**Keywords:** family financial stress, acculturation, emotional and behavioral problems, psychological distress, parenting styles, refugee children

## Abstract

The present study evaluated the application of the basic and extended (incorporated primary caregivers’ levels of acculturation) Family Stress Model (FSM) to understand the effect of family financial stress and primary caregivers’ levels of acculturation on children’s emotional and behavioral problems among refugees in Australia. A total of 658 refugee children aged 5–17 and their primary caregivers (*n* = 410) from the third wave of a nationwide longitudinal project were included in this study. We used multilevel structural equation models with bootstrapping to test the indirect effects of family financial stress and caregivers’ levels of acculturation (including English proficiency, self-sufficiency, social interaction, and self-identity) on children’s emotional and behavioral problems through caregivers’ psychological distress and parenting styles. The results showed that the extended FSM improved the model fit statistics, explaining 45.8% variation in children’s emotional and behavioral problems. Family financial stress, caregivers’ English proficiency, and self-identity had indirect effects on children’s emotional and behavioral problems through caregivers’ psychological distress and hostile parenting. The findings showed that interventions aimed at reducing caregivers’ psychological distress and negative parenting could be effective in alleviating the adverse effects of family financial stress and caregivers’ low levels of acculturation on refugee children’s mental health.

## 1. Introduction

By the end of 2018, the number of forcibly displaced population increased to 70.8 million globally, including 25.9 million refugees who needed to resettle in a third country [1]. About half of the refugee population were children under 18 years old [1]. Due to a variety of stressors associated with forced migration, the available evidence suggests that refugee children are more vulnerable to mental health problems, including posttraumatic stress disorders, depression, anxiety disorders, and emotional and behavioral problems than children in the host population [2,3]. Childhood emotional and behavioral problems are often the early manifestation of adult mental disorders, which have a profound impact on personality formation, social function, and lifelong achievements [4,5]. 

Evidence from previous research shows that children’s emotional and behavioral problems are associated with a myriad of family stressors, including low socioeconomic status [6], tense family relationships [7,8], and parents’ mental health problems [9,10]. The Family Stress Model (FSM) [11] is a theoretical framework commonly used to study how family financial stress affects children’s adjustment problems. The FSM has been widely applied in diverse ethnic groups [12,13], including Hispanics, African Americans, European Americans, and Mexican Americans. These studies reported consistently that family financial stress exacerbates children’s adjustment problems through parental psychological distress and disrupted parenting (Figure 1). The three core dimensions under the FSM, namely family financial stress, parents’ mental health problems, and disrupted parenting, are common family stressors among the refugee population. Therefore, we chose the FSM as a theoretical framework to guide this study. 

However, the FSM was first proposed based on general populations (Midwestern U.S. farm families) [11] and has not been operationalized to date among refugee populations. Existing literature shows that, in addition to the above stressors covered in the FSM, the primary caregivers’ levels of acculturation is also a very important family stressor among refugees [14,15]. Therefore, we assumed that extending the FSM by incorporating primary caregivers’ levels of acculturation could improve the applicability of the FSM to the refugee population. Furthermore, by using the extended FSM, we could explore the relationships between primary caregivers’ levels of acculturation and children’s mental health among refugee populations, which has not been studied before.

Broadly, acculturation refers to the process of culture change and adaptation when individuals or groups from different cultural backgrounds come into contact with each other [16]. In this process, refugees may experience major life changes that lead to the deterioration of mental health and changes in parenting styles, which are closely related to children’s mental health [17,18]. However, identifying the association between primary caregivers’ levels of acculturation and children’s mental health is challenging, because a balance between what needs to be measured and what can be measured in epidemiological studies and health surveys is controversial [19]. 

Acculturation was originally described as a unidimensional process in which individuals or groups were assumed to acquire the culture of the host country while relinquishing their heritage culture [20]. However, Berry [21] put forward a bi-dimensional acculturation theory in which retention of the heritage culture occurred independently of acquisition of the host culture, and acculturation could be divided into four categories: assimilation, separation, integration, and marginalization. Although there was evidence to support the bi-dimensional acculturation theory [22], most self-reported acculturation scales were based on unidimensional acculturation theory in public health studies [23], and most studies showed that a linear relationship between acculturation and psychological distress was well demonstrated by the unidimensional theory [19]. The only one study on the relationship between acculturation and mental health among refugees in Australia adopted the bi-dimensional acculturation theory [24]. The study showed that levels of acculturation of host country is negatively associated with mental health problems, while the relationship between levels of acculturation of heritage culture and mental health problems was not significant. 

This study has two purposes: (1) to extend the FSM by testing whether primary caregivers’ levels of acculturation have an impact on children’s emotional and behavioral problems and (2) to evaluate the application of the basic and extended FSM to the refugee population in Australia. 

## 2. Methods 

### 2.1. Data Sources

The third wave of data from the “Building a New Life in Australia” (BNLA) project was used in this study [25]. BNLA is the first national cohort study documenting how humanitarian migrants (HMs) settle into life in Australia. The BNLA project is funded by the Australian Department of Social Services (DSS) and is managed by the Australian Institute of Family Studies (AIFS). Data in the third wave of data were collected between October 2015 and February 2016 via face-to-face interviews. The questionnaire was translated into nine languages to cover participants from diverse cultural and linguistic backgrounds. Computer-Assisted Self-Interview (CASI) was a frontline interview method. If CASI failed, participants could still complete the questionnaire through Computer-Assisted Personal Interview, and an accredited interpreter assisted with the interview. Details about the BNLA project are provided by the Australian Department of Social Services (https://www.dss.gov.au/settlement-services-publications/building-a-new-life-in-australia-bnla-the-longitudinal-study-of-humanitarian-migrants). 

### 2.2. Sampling and Study Participants

BNLA participants consisted of principal applicants (PAs) and secondary applicants (SAs) for a permanent humanitarian visa in Australia. A PA is the main applicant in a migrating unit (MU), and SAs are other members of the MU. Participants were recruited from 11 sites where most HMs settled, covering five major cities of Sydney, Melbourne, Brisbane, Adelaide, and Perth and another six regional areas across Australia. Within the 11 study sites, all PAs aged 18 years and older, who had been granted a permanent humanitarian visa between May and December 2013 were contacted to take part in the BNLA project. Once the PAs agreed to join the project, SAs in the same MUs were also invited to participate. In total, 1509 MUs and 2399 individuals within these MUs were included in the BNLA project, and annual follow-up surveys have been conducted since 2013. 

Unlike the first and second waves, the third wave included a module focused on children aged 5–17 years. The child module covered the Strengths and Difficulties Questionnaire (SDQ) [26], health status, trauma, and how children were adjusting to life in Australia. Primary caregivers were invited to complete the questionnaire for all children aged 5–17 years, and in addition those children aged 11–17 years completed a self-reported questionnaire to check that self-completed scores did not differ from scores obtained from caregivers’ completed questionnaire. The criteria for selecting children in each MU was mainly as follows: (1) up to two children per household could participate and (2) children had to be aged between 5 and 17 years. Two children were randomly selected if there were three or more children between 11 and 17 years in the selected MU. Where there was only one child aged 11–17 years and several children aged 5–10 years in the household, the one child older than 11 years was selected and one child aged 5–10 years was randomly selected; whereas in households with only children aged 5–10 years, up to two children were randomly selected.

### 2.3. Measures

Questions for all study variables and their rating scales are summarized in Table 1.

#### 2.3.1. Children’s Emotional and Behavioral Problems

Children’s emotional and behavioral problems were measured by the Strengths and Difficulties Questionnaire (SDQ) total difficulties score [26]. The SDQ has good reliability and validity, and it has been extensively used to screen for child and adolescent psychiatric disorders in many countries, including Australia [3,27,28]. The SDQ consists of 25 items, and each item is rated on a 3-point scale (0 = not true, 1 = somewhat true, 2 = certainly true). The 25 items are distributed into five domains, namely emotional symptoms, conduct problems, hyperactivity, peer problems, and prosocial behavior. Each domain has five items and the scores are generated by summing scores of the five items, giving a total score for each domain that ranges from 0 to 10. A final total scale score is generated by summing scores of the four domains (i.e., emotional symptoms, conduct problems, hyperactivity, and peer problems scales) [26], which range from 0 to 40, with higher scores indicating a higher risk of emotional and behavioral problems. The internal consistency reliability of the SDQ total difficulties score in this study was 0.71.

Although the SDQ questionnaire was completed by primary caregivers of children aged 5–17 years and self-completed among those children aged 11–17 years, there was no statistically significant difference between the SDQ total difficulties scores reported by primary caregivers and children themselves (*p* = 0.742) (see Appendix A
Table A1). To ensure consistency of the SDQ total difficulties score of children under 11 years and children aged 11 years or older, we used the score provided by primary caregivers for both age groups.

#### 2.3.2. Family Financial Stress

Family financial stress was measured by six items that asked the participants whether they were unable to pay for daily necessities and needed financial help due to lack of money in the last 12 months. These were (1) could not pay gas, electricity, or telephone bills; (2) could not pay the rent or mortgage payments; (3) went without meals; (4) were unable to heat or cool their home; (5) pawned or sold something because they needed cash; (6) needed help from a welfare or community organization. The sum of the scores (0 = no, 1 = yes) of the six items represented the number of financial stress indicators (ranging from 0 to 6), with a higher score reflecting greater financial stress.

#### 2.3.3. Primary Caregivers’ Acculturation Level

The BNLA’s acculturation measure included a whole range of dimensions encompassing English proficiency, self-sufficiency to participate in life in Australia, social interaction, and self-identity at the primary caregivers’ level, which is consistent with our previous research [29]. “English proficiency” was measured by four items, including the abilities to understand spoken English, speak English, read English, and write English. Primary caregivers were asked to rate each item on a 4-point scale (1 = not at all to 4 =very well). A higher score reflected a higher level of English proficiency. “Self-sufficiency to participate in life in Australia” was measured by eight items, primary caregivers were asked if they knew how to look for a job, find a school or child care for children, use public transport, get help in an emergency, use bank services, find out what government services and benefits are available, find out about their rights, get help from the police. Each item was scored on a 4-point scale (1 = would not know at all to 4 = would know very well). A higher score reflected a higher level of self-sufficiency. “Social interaction” was measured by three items, including how easy it was to make friends, understand Australian culture, and talk to neighbors. Each item was scored on a 4-point scale (1 = very hard to 4 = very easy). A higher score reflected a higher level of social interaction. “Self-identity” was measured by three items regarding feeling part of the Australian community (1 = hardly ever to 4 = always), feeling welcomed in Australia (1 = hardly ever to 4 = always) and experiencing discrimination in Australia (0 = yes; 1 = no). A higher score reflected a higher level of self-identity. The internal consistency reliability of English proficiency, self-sufficiency to participate in life in Australia, social interaction, and self-identity was 0.94, 0.92, 0.80, and 0.54, respectively, and the internal consistency reliability of the total acculturation was 0.91.

#### 2.3.4. Primary Caregivers’ Psychological Distress

Primary caregivers’ psychological distress was assessed by the 6-item Kessler psychological distress scale (K6) [30]. The K6 has been translated into different languages for screening psychological distress cross-culturally [31]. Each item was scored on a 5-point scale (1 = none of the time to 5 = all of the time). The Australian K6 total score was calculated by summing up scores of each item (ranging from 6 to 30), with higher scores indicating a higher risk of psychological distress. In this study, the reliability test showed strong internal consistency with a Cronbach’s alpha of 0.90.

#### 2.3.5. Parenting Style

Parenting style consisted of two dimensions: warm parenting and hostile parenting. The warmth sub-scale was used to assess warm parenting and derived from the Child Rearing Questionnaire [32]. The sub-scale included five items that indicated positive emotional expression by primary caregivers towards their children. Each item was scored on a 5-point scale (1 = none of the time to 5 = all of the time). The hostility sub-scale was used to assess hostile parenting and derived from the Early Childhood Longitudinal Study of Children [33]. The sub-scale included five items that indicated negative expression and behavior by primary caregivers toward their children. Each item was scored on a 5-point scale (1 = rarely to 5 = almost always). The internal consistency reliability of the warmth scale and the hostility scale was 0.75 and 0.73, respectively.

#### 2.3.6. Covariates

In this study, we included primary caregivers’ and children’s demographics as covariates, including children’s age, gender, physical health (1 = poor to 5 = excellent), achievement at school (1 = below average to 4 = excellent), experiences of pre-migration trauma (0 = yes; 1 = no), household structure (0 = single parent family; 1 = couple family), and primary caregivers’ age, gender and education (1 = never attended school to 5 = university degree).

### 2.4. Statistical Analyses

Analyses were conducted in IBM SPSS 25.0, Mplus 7.4 (Muthen & Muthen, Los Angeles, CA, USA) and R 3.6.1. Descriptive statistics including the mean, standard deviation (SD), frequency, and proportion were used to describe all study variables and participants’ sociodemographic characteristics. Cronbach’s alpha coefficients were used to test the internal consistency of each latent variable. Correlation analysis was used to describe the correlations among all constructs. Correlation analysis results (see Appendix A
Table A2) showed that warm parenting was not related to parental psychological distress and children’s emotional and behavioral problems; correspondingly, all pathways related to warm parenting were deleted in the subsequent multilevel structural equation modeling analyses.

Clustered data (children level is level-1, and the primary caregivers’ level is level-2) were analyzed using Multilevel Structural Equation Models (MSEM) [34]. Consistent with the basic FSM, at level-1, there were two latent variables, namely children’s emotional and behavioral problems and hostile parenting. At level-2, there were four latent variables in the extended FSM, namely English proficiency, self-sufficiency, social interaction, and self-identity, and two observed variables, i.e., family financial stress and primary caregivers’ psychological distress. Due to non-normality and non-independence in the data, we used the MSEM with robust maximum-likelihood estimation to test complex relationships between studies variables in the basic FSM as well as in the extended FSM and the bootstrapping method (i.e., 5000 bootstrap samples were used in R 3.6.1) to test the indirect effects of financial stress, English proficiency, self-sufficiency, social interaction, and self-identity on children’s emotional and behavioral problems through primary caregivers’ psychological distress and hostile parenting. Using bootstrapping addresses the lack of normality and provides stronger accuracy of intervals [35]. All covariates of level-1, and significant covariates of level-2 were adjusted. The data-model fitting effect was assessed using the following indicators: Comparative Fit Index (CFI) (≥0.90), Tucker–Lewis index (TLI) (≥0.90), Standardized Root Mean Square Residual (SRMR) (≤0.08), Root Mean Square Error of Approximation (RMSEA) (≤0.08), and the normed chi-square ($\chi^{2}$/*d.f.*) (≤3.0). Missing data, except the outcome variable (the SDQ total difficulties score), were processed using the Mplus default, full-information maximum-likelihood (FIML). In terms of the SDQ total difficulties score, we firstly excluded individuals with missing data on this variable in the main models, then sensitivity analyses were performed to compare results of models with and without missing data, while the missing data were also processed using the FIML.

### 2.5. Ethics

The BNLA project data are publicly available to approved researchers (W.C. and A.M.N.R.). The project was conducted in accordance with the Declaration of Helsinki and was approved by the Ethics Committee of Western Sydney University’s Human Research Ethics (no. EX2016/01). All participants and (if the participants were under 18 years of age) their parents were informed about the project and all of them provided informed consent. 

## 3. Results

Of 1894 participants in the third wave of BNLA program, 425 were eligible primary caregivers who have children aged 5–17 years old. Among the 425 primary caregivers, fifteen (3.5%) were excluded from this study because they did not report the SDQ total difficulties scores for their children. A total of 410 eligible primary caregivers were included in the study completing questionnaire surveys for 658 refugee children. 

### 3.1. Descriptives

Table 1 and Table 2 present descriptive statistics for all study variables and participants’ sociodemographic characteristics. The mean SDQ total difficulties score was 9.03 (SD = 5.60), and the mean scores of emotional symptoms, conduct problems, hyperactivity, and peer problems were 2.24 (SD = 2.25), 1.38 (SD = 1.57), 3.02 (SD = 2.19), and 2.40 (SD = 1.55), respectively. Among the 658 children, slightly more than half of them (53.8%) were male, and the mean age was 12.0 years (SD = 3.5). Most children (92.5%) reported good physical condition, and 54.4% of them had above average achievement at school. There were 189 (28.7%) children who experienced or witnessed trauma before coming to Australia. Of the 410 primary caregivers, most of them (76.1%) were female, and the mean age was 40.1 years (SD = 8.8). There were 93 (22.7%) primary caregivers who never attended school, eighty (19.8%) had received less than six years of schooling, and 57.3% of them had received more than seven years of schooling.

### 3.2. Multilevel Structural Equation Modeling Analysis

The basic FSM (Figure 2) was marginally fitted to the data, as shown in Table 3. The normed chi-square value ($\chi^{2}$/d.f.) was below 3.0, RMSEA and SRMR were both below 0.08, but CFI and TLI were marginally below 0.90. However, the extended FSM (Figure 3) fitted the data well, as shown in Table 3. The normed chi-square value ($\chi^{2}$/d.f.) was below 3.0, CFI and TLI were above 0.90, and RMSEA and SRMR were also below 0.08. 

Figure 3 showed that hostile parenting (β = 0.595, *p* < 0.001) and primary caregivers’ psychological distress (β = 0.236, *p* = 0.003) had direct effects on children’s emotional and behavioral problems, with the model explaing 45.8% of the variance in children’s emotional and behavioral problems. However, family financial stress, primary caregivers’ English proficiency, self-sufficiency, social interaction, and self-identity had no significant direct effects on children’s emotional and behavioral problems. Nonetheless, family financial stress, primary caregivers’ English proficiency, and self-identity were found to indirectly impact children’s emotional and behavioral problems through primary caregivers’ psychological distress (Table 4). We tested all indirect effects by bootstrapping of the sample. The indirect effect of financial stress on children’s emotional and behavioral problems occurred through primary caregivers’ psychological distress (β = 0.061, 95% CI: (0.016, 0.119)) and through the mediating chain consisting of psychological distress and hostile parenting (β = 0.021, 95% CI: (0.001, 0.053)). Similarly, the indirect effect of self-identity on children’s emotional and behavioral problems occurred through primary caregivers’ psychological distress (β = −0.141, 95% CI: (−0.312, −0.023) and through the mediating chain consisting of psychological distress and hostile parenting (β = −0.049, 95% CI: (−0.136, −0.001)). In addition, the indirect effect of English proficiency on children’s emotional and behavioral problems occurred through primary caregivers’ psychological distress (β = −0.067, 95% CI: (−0.165, −0.001)) (all indirect effects were unstandardized).

Additionally, Figure 3 presented that financial stress (β = 0.283, *p* < 0.001), English proficiency (β = −0.122, *p* = 0.041), and self-identity (β = −0.191, *p* = 0.008) directly impacted on primary caregivers’ psychological distress, and primary caregivers’ psychological distress directly impacted on hostile parenting (β = 0.139, *p* = 0.045).

### 3.3. Sensitivity Analysis

We included the individuals with missing the outcome variable (i.e., the SDQ total difficulties score) for sensitivity analysis to ascertain the robustness of the results, and the sensitivity analysis produced similar results (see Appendix A
Figure A1, Figure A2 and Table A3) to those from our main models.

## 4. Discussion

The results suggest that the basic FSM was marginally applicable to the sample drawn from the refugee population in Australia. The extended FSM simultaneously examined the effects of two common stressors of refugees, i.e., family financial stress and primary caregivers’ levels of acculturation, on refugee children’s emotional and behavioral problems, and the extended FSM improved the model fit statistics and fitted the data well.

First, we found that hostile parenting and primary caregivers’ psychological distress both had a direct effect on children’s emotional and behavioral problems. This is consistent with previous studies [7,36] showing that primary caregivers’ psychological distress and negative parenting were associated with an increase in children’s mental health problems. Furthermore, these studies showed that primary caregivers’ psychological distress and negative parenting often coexisted with decreased parent–child interaction, weakened sense of family connection, conflict among family members, and other adverse family environments or functions, which were closely related to children’s mental health problems.

Family financial stress was found to indirectly impact children’s emotional and behavioral problems through primary caregivers’ psychological distress. Family financial stress exacerbated primary caregivers’ psychological distress and the latter led to increases in emotional and behavioral problems among refugee children. Consistent with findings in the UK and USA [37,38], primary caregivers’ psychological distress had a striking effect on the relationship between family financial stress and child mental health problems. Furthermore, caregivers’ psychological distress may filter into parenting styles, which tend to be negative parenting [39]. This likely plausible as our findings found that primary caregivers’ psychological distress led to hostile parenting, and in turn, subsequent emotional and behavioral problems in children. Altogether, these findings are consistent with other studies among the general children population based on the FSM [40].

Among the four dimensions of acculturation measured in this study, primary caregivers’ English proficiency and self-identity showed significant indirect effects on refugee children’s emotional and behavioral problems. Consistent with previous research [18,41], we found that children of primary caregivers with low levels of self-identity were more likely to suffer emotional and behavioral problems. Higher levels of self-identity may be linked to more social ties with natives and higher willingness to engage in local social activities [42]. 

English proficiency was found to be an important domain in acculturation among refugees [43,44], however there was no direct effect of caregivers’ English proficiency on children’s emotional and behavioral problems in this study. This result is consistent with findings by Chen et al., in which children’s psychological adjustment was more directly affected by their English proficiency rather than caregivers’ English proficiency [45]. However, the indirect effect of caregivers’ English proficiency on children’s emotional and behavioral problems through caregivers’ psychological distress was significant. This is consistent with previous studies, in which higher levels of caregivers’ English proficiency were associated with better mental health of their children [46,47]. It is worth noting that we found English proficiency of refugee parents to be still generally poor (mean scores were around 2 (not well)) in the third round of survey of the BNLA project, which was about two and a half years of resettlement. The language barrier leads to social withdrawal, difficulty with service utilization, and more discrimination [48,49], and in turn is associated with greater psychological distress. From the perspective of relieving refugees’ psychological distress, the findings suggest that it is necessary to enhance the availability of language assistance services and the effectiveness of existing English training services, e.g., the Adult Migrant English Program (AMEP) [50], and to provide long-term language training to refugees.

Our findings suggest that strategies to strengthen family functions by reducing primary caregivers’ psychological distress and negative parenting could be effective in alleviating the adverse effects of financial stress and low levels of primary caregivers’ acculturation on refugee children’s emotional and behavioral problems. The potential measures include psychosocial interventions [51], stress management exercises, and parenting interventions, e.g., Parenting Management Training [52]. Additionally, providing settlement services that directly reduce financial stress and improve acculturation levels, such as transitional housing, employment assistance, job training, social activities in multicultural communities, language training, and assistance, is also important in promoting refugee adults’ and children’s mental health. 

Furthermore, our findings suggest that the extended FSM incorporating primary caregivers’ levels of acculturation can be used to better illustrate the relationships between parental post-migration stressors and children’s mental health among refugee populations. The measurement of acculturation in this study is based on the unidimensional theory, which mainly reflects the acquisition of host culture. In view that the theory of acculturation is still controversial [53], in future research, acculturation can be measured from the two dimensions of retaining heritage culture and acquiring host culture based on the bi-dimensional theory, and the effects of the four categories of acculturation (including assimilation, separation, integration, and marginalization) on mental health of refugees and their children can be explored.

This study has several limitations. First, due to limited information in the questionnaire of the BNLA project, we could not measure acculturation by a standard scale and did not control the impact of children’s acculturation on their emotional and behavioral problems, which may limit the comparability between the current study and others. Second, other aspects of parenting style, e.g., inconsistent parenting, were not measured in this study, which could be further assessed in the future. Third, self-reported data were used in this study, therefore the accuracy of results might be affected by information bias. Finally, because of the cross-sectional nature, causal relationships cannot be inferred from our findings in this study.

## 5. Conclusions

The extended FSM was more applicable to the refugee population in Australia than the basic FSM. Primary caregivers’ levels of acculturation were an important component of the extended FSM which indirectly affected children’s emotional and behavioral problems. The findings of this study provide evidence that caregivers’ psychological distress and hostile parenting are directly associated with children’s emotional and behavioral problems, and family financial stress, caregivers’ English proficiency, and self-identity have indirect effects on refugee children’s emotional and behavioral problems through primary caregivers’ psychological distress and hostile parenting. Our findings suggest that interventions and services aimed at reducing financial stress and improving acculturation levels have potential benefits for both refugee adults and their children. More importantly, interventions that reduce primary caregivers’ psychological distress and negative parenting could be effective in alleviating the adverse effects of family financial stress and low levels of primary caregivers’ acculturation on refugee children’s mental health.

## Figures and Tables

**Figure 1 ijerph-17-02716-f001:**
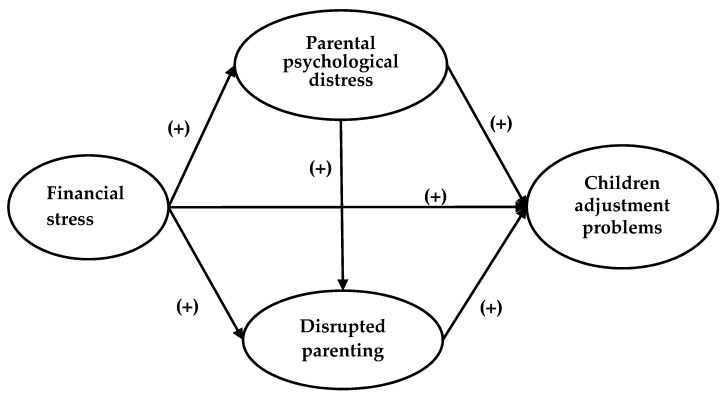
The Family Stress Model framework.

**Figure 2 ijerph-17-02716-f002:**
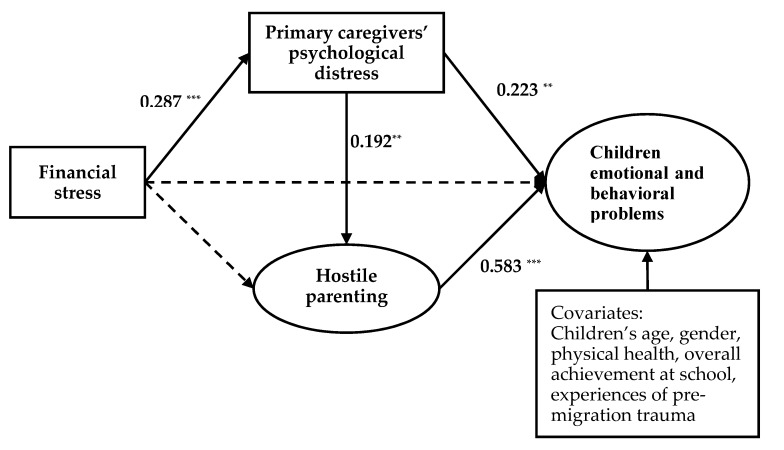
Basic Family Stress Model (FSM). Note. ** *p* < 0.01, *** *p* < 0.001. Coefficients are standardized path coefficients. The solid lines represent significant paths (*p* < 0.05). The dashed lines represent non-significant paths. Children’s age, gender, physical health, overall achievement at school, and experiences of pre-migration trauma are included as covariates.

**Figure 3 ijerph-17-02716-f003:**
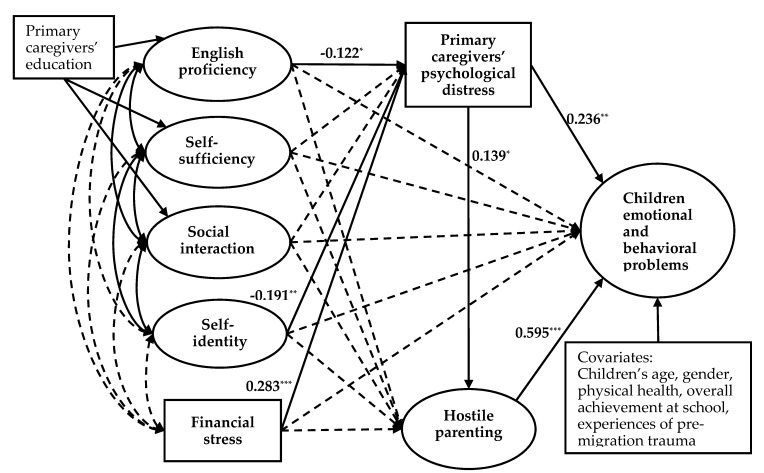
Extended FSM including primary caregivers’ levels of acculturation. Note. * *p* < 0.05, ** *p* < 0.01, *** *p* < 0.001. Coefficients are standardized path coefficients. The solid lines represent significant paths (*p* < 0.05). The dashed lines represent non-significant paths. Children’s age, gender, physical health, overall achievement at school, experiences of pre-migration trauma, and primary caregivers’ education are included as covariates.

**Table 1 ijerph-17-02716-t001:** Descriptive statistics for all study variables.

Variables	Mean (SD)/n (%)	Cronbach’s α	Range/Valuation	Missing Value/n
*Level-1 (n = 658)*				
**Children’s emotional and behavioral problems**		0.71		
SDQ total difficulties score	9.03 (5.60)		0–29	0
SDQ emotional symptoms score	2.24 (2.25)		0–10	0
SDQ conduct problems score	1.38 (1.57)		0–10	0
SDQ hyperactivity score	3.02 (2.19)		0–10	0
SDQ peer problems score	2.40 (1.55)		0–8	0
**Parenting style**				
** Warm parenting**		0.75		
Has warm close times with child	4.34 (0.88)		1 = never/almost never; 2 = rarely; 3 = sometimes; 4 = often; 5 = always/almost always.	17
Enjoys doing things with and listening to child	4.33 (0.86)			22
Talk about what is going on in their life	3.56 (1.30)			34
Is good at getting child to do what is told	3.75 (1.20)			33
Feels close to child when child is happy and upset	4.21 (1.07)			30
** Hostile parenting**		0.73		
Has been angry with child	2.05 (1.06)		1 = never/almost never; 2 = rarely; 3 = sometimes; 4 = often; 5 = always/almost always.	28
Has raised voice or shouted at child	1.98 (0.97)			28
Gets on my nerves when child cries	2.28 (1.35)			38
Has lost temper with child	1.58 (0.93)			27
Has left child alone in bedroom when upset	1.44 (0.92)			29
*Level-2 (n = 410)*				
**Primary caregivers’ acculturation level**		0.91		
** English proficiency**		0.94		
Understand spoken English	2.20 (0.71)		1 = not at all; 2 = not well; 3 = well; 4 = very well.	11
Speak English	2.04 (0.74)			6
Read English	2.15 (0.82)			7
Write English	2.07 (0.77)			8
**Self-sufficiency to participate in the life in Australia**		0.92		
Look for a job	1.76 (0.92)		1 = would not know at all; 2 = would know a little; 3 = would know fairly well; 4 = would know very well.	4
Find a school or childcare for children	2.31 (1.04)			22
Use public transport	2.93 (1.04)			4
Get help in an emergency	2.68 (1.06)			3
Use bank services	2.23 (1.08)			4
Find out what government services and benefits are available	2.30 (1.04)			4
Find out about your rights	2.24 (1.05)			4
Get help from the police	2.59 (1.09)			3
** Social interaction**		0.80		
Make friends in Australia	2.51 (0.76)		1 = very hard; 2 = hard; 3 = easy; 4 = very easy.	23
Understand Australian ways/culture	2.53 (0.76)			20
Talk to Australian neighbors	2.30 (0.81)			27
** Self-identity**		0.54		
Feel welcome in Australia	3.57 (0.67)		1 = hardly ever; 2 = some of the time; 3 = most of the time; 4 = always.	3
Feel part of the Australian community	3.38 (0.86)			2
Experience discrimination in Australia n (%)				2
Yes	24 (5.85)			
No	384 (93.66)			
**Financial stress**	1.22 (1.51)		0–6	5
**Psychological distress**	14.31 (6.01)	0.90	6–30	8

SDQ: Strengths and Difficulties Questionnaire.

**Table 2 ijerph-17-02716-t002:** Participants’ characteristics.

Characteristics	*n* (%)	Mean (SD)
**Primary caregivers’ characteristics (*n* = 410)**		
Age (years)		40.1 (8.8)
Gender		
Male	98 (23.9)	
Female	312 (76.1)	
Years of education		
Never attended school	93 (22.7)	
≤6 years of schooling	80 (19.8)	
≥7 years of schooling	175 (43.2)	
Trade or technical qualification beyond school	21 (5.2)	
University degree	36 (8.9)	
Household structure		
Single parent family	103 (25.1)	
Couple family	301 (73.4)	
**Children’s characteristics (*n* = 658)**		
Age (years)		12.0 (3.5)
Gender		
Male	354 (53.8)	
Female	304 (46.2)	
Physical health		4.1 (1.0)
Poor	16 (2.4)	
Fair	31 (4.7)	
Good	125 (19.0)	
Very good	218 (33.1)	
Excellent	266 (40.4)	
Overall achievement at school		3.8 (1.0)
Below average	39 (5.9)	
Average	252 (38.3)	
Above average	172 (26.1)	
Excellent	186 (28.3)	
Experiences of pre-migration trauma		
Yes	189 (28.7)	
No	457 (69.5)	

**Table 3 ijerph-17-02716-t003:** Goodness-of-fit test for the basic and extended FSM model.

Model	*χ* ^2^	*d.f.*	*χ*^2^/*d.f.*	CFI	TLI	RMSEA	SRMR
Basic FSM	242.442	105	2.309	0.880	0.844	0.045	0.079
Extended FSM	751.0455	437	1.720	0.939	0.928	0.034	0.062

Note: d.f. = Degrees of Freedom; CFI = Comparative Fit Index; TLI = Tucker–Lewis Index; RMSEA = Root Mean Square Error of Approximation; SRMR = Standardized Root Mean Square Residual.

**Table 4 ijerph-17-02716-t004:** The indirect effects of financial stress, English proficiency, self-sufficiency, social interaction, and self-identity on children’s emotional and behavioral problems in the Multilevel Structural Equations Model (MSEM).

Pathways	β	95% CI	
		Lower	Upper
**Extended FSM**			
** Family financial stress**			
a. Financial stress → Psychological distress → Hostile parenting → Children’s emotional and behavioral problems	0.021	0.001	0.053
b. Financial stress → Psychological distress → Children’s emotional and behavioral problems	0.061	0.016	0.119
c. Financial stress → Hostile parenting → Children’s emotional and behavioral problems	0.037	−0.038	0.125
** English proficiency**			
a. English proficiency → Psychological distress → Hostile parenting → Children’s emotional and behavioral problems	−0.023	−0.071	0.002
b. English proficiency → Psychological distress → Children’s emotional and behavioral problems	−0.067	−0.165	−0.001
c. English proficiency → Hostile parenting → Children’s emotional and behavioral problems	−0.034	−0.238	0.165
** Self-sufficiency**			
a. Self-sufficiency → Psychological distress → Hostile parenting → Children’s emotional and behavioral problems	0.014	−0.020	0.065
b. Self-sufficiency → Psychological distress → Children’s emotional and behavioral problems	0.039	−0.053	0.153
c. Self-sufficiency → Hostile parenting → Children’s emotional and behavioral problems	0.058	−0.184	0.324
** Social interaction**			
a. Social interaction → Psychological distress → Hostile parenting → Children’s emotional and behavioral problems	−0.023	−0.097	0.026
b. Social interaction → Psychological distress → Children’s emotional and behavioral problems	−0.066	−0.236	0.068
c. Social interaction → Hostile parenting → Children’s emotional and behavioral problems	−0.225	−0.603	0.094
** Self-identity**			
a. Self-identity → Psychological distress → Hostile parenting → Children’s emotional and behavioral problems	−0.049	−0.136	−0.001
b. Self-identity → Psychological distress → Children’s emotional and behavioral problems	−0.141	−0.312	−0.023
c. Self-identity → Hostile parenting → Children’s emotional and behavioral problems	−0.284	−0.675	0.035

Note: MSEM: Multilevel Structural Equation Model; β = unstandardized indirect effects; 95% CI = 95% Confidence Interval.

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
