# Peer review of "The Effects of Family Financial Stress and Primary Caregivers’ Levels of Acculturation on Children’s Emotional and Behavioral Problems among Humanitarian Refugees in Australia"

_ijerph, 2020, doi:10.3390/ijerph17082716_

Round 1

Reviewer 1 Report

This well presented study throws new light on the experience of refugees in Australia. It focuses on parental stress factors affecting the well being of children in refugee households. It is innovative insofar as it adds a 'child module' in the third wave of family research. Whilst the authors define the concept of acculturation, they do not sufficiently reflect that this is a contested concept. They focus on the refugees' acculturation but do not elaborate on the receiving country's preparedness to accept refugees or minorities. Such issues are explored in the following publication, which the authors would benefit from incorporating in their study. This would provide necessary balance: Am Psychol. Author manuscript; available in PMC 2013 Jul 3. Published in final edited form as: Am Psychol. 2010 May-Jun; 65(4): 237–251. doi: 10.1037/a0019330 Rethinking the Concept of Acculturation - Implications for Theory and Research Seth J. Schwartz, Jennifer B. Unger, Byron L. Zamboanga, and José Szapocznik

Reviewer 2 Report

Dear Authors,

Congratulation for your work!

It is an interesting article about how FSM research paradigm is applicable to the refugee population in Australia. It seems that caregivers’ levels of acculturation may be an important new component of the FSM because it indirectly affects children’s emotional and behavioral problems.

I have two suggestions to improve the

Introduction part:

  • Please underlie why you choose FSM as the theoretical paradigm and despite its strong empirical what gap in the literature covers your study.
  • Please offer the conceptualization of acculturation you follow. Do you know other studies that have been done on refugee population’s acculturation in Australia?

Methodological part:

  • Please describe how you dealt with missing data. Do you have missing data? How many?

Discussion part

  • Please stress the recommendations for future research regarding the acculturation as a component of FSM

Reviewer 3 Report

In “Table A1. Differences in total and subscale SDQ scores reported by primary caregivers and refugee children (MEAN (SD))” it only appears a value, is it t or p? It is important to include both values or you can either include this information in notes below de table. In case of referring to p  values, you should include this information in lines 145-147.

You must erase the information below “Table 3. Goodness-of-fit test for the basic and extended FSM model” related to acceptable ranges and values of goodness of fit indexes because it has been already mentioned in other parts of the manuscript.

You should write P in lower case.
